# Characterization of Iflavirus in the Red Flour Beetle, *Tribolium castaneum* (Coleoptera; Tenebrionidae)

**DOI:** 10.3390/insects14030220

**Published:** 2023-02-23

**Authors:** Soheila Fatehi, Michael Aikins, Thomas W. Philips, Susan Brown, Kun Yan Zhu, Erin D. Scully, Yoonseong Park

**Affiliations:** 1Department of Entomology, Kansas State University, Manhattan, KS 66506, USA; 2Division of Biology, Kansas State University, Manhattan, KS 66506, USA; 3Stored Product Insect and Engineering Research Unit, USDA-ARS-CGAHR, Manhattan, KS 66502, USA

**Keywords:** iflavirus, *Tribolium castaneum*, *TcIV*, virus, beetle

## Abstract

**Simple Summary:**

A model beetle, *Tribolium castaneum*, carries a positive single-stranded RNA virus *Tribolium castaneum iflavirus* (*TcIV*). This study shows that the *TcIV* is prevalent among different laboratory cultures, different strains, and the same strain maintained in different laboratories, and in the *Tribolium* cell line TcA. Transovarial transmissions and their prevalence in the nervous system were supported in this study. The novel *TcIV*, showing no observable pathogenicity, offers an opportunity to study the interaction between the virus and the immune system in this model beetle species.

**Abstract:**

Iflavirus is a group of viruses distributed mainly in arthropod species. We surveyed *Tribolium castaneum iflavirus* (*TcIV*) in different laboratory strains and in Sequence Read Archives (SRA) in GenBank. *TcIV* is highly specific to only *T. castaneum* and is not found in seven other Tenebrionid species, including the closely related species *T. freemani*. The same strains from different laboratories and different strains displayed largely different degrees of infections in the examination of 50 different lines by using Taqman-based quantitative PCR. We found that ~63% (27 out of 43 strains) of *T. castaneum* strains in different laboratories are positive for *TcIV* PCR with large degrees of variation, in the range of seven orders of magnitude, indicating that the *TcIV* is highly fluctuating depending on the rearing conditions. The *TcIV* was prevalent in the nervous system with low levels found in the gonad and gut. The transovarial transmission was supported in the experiment with surface-sterilized eggs. Interestingly, *TcIV* infection did not show observable pathogenicity. *TcIV* offers an opportunity to study the interaction between the virus and the immune system of this model beetle species.

## 1. Introduction

The red flour beetle, *Tribolium castaneum*, is a model organism and economically important insect pest with numerous molecular tools, such as a high-quality genome assembled at the chromosomal level, highly efficient systemic RNA interference, cell line, and CRISPR/cas9-mediated transgenesis [1,2,3,4,5,6,7,8] and widely utilization in various studies including the insect immune system [4]. Iflavirus in *T. castaneum* was first reported by Lewis et al. in 2018 [9] by a partial sequence identified by studying the immune system mediated through small RNAs against viruses. Iflaviruses (Picornavirales; Iflaviridae) are characterized by a positive single-stranded RNA genome with structural proteins in the N-terminal end of the viral genome and non-structural proteins at the C-terminus. The name iflavirus has been derived from infectious flacherie virus (IFV), the cause of infectious flacherie disease, and also from the cocoon loss of *Bombyx mori* in Japan, which was the first iflavirus described and sequenced [10]. So far, 24 species of iflaviruses have been recognized based on the NCBI (National Center for Biotechnology Information) virus genomes for discovery as of November 2022.

The most-studied member of the genus *Iflavirus*, the deformed wing virus (DWV), is the causative agent for honeybee hive collapsing syndrome, which is transmitted by the parasitic mite *Varroa destructor* [11,12,13,14,15,16,17,18] Infection by iflavirus generally occurs orally by feeding on contaminated food, and the virus appears to be distributed in the gut, reproductive organs, fat body, muscle, and central nervous system (CNS) [19]. Iflaviruses are believed to be arthropod-specific viruses that infect insects in the orders Lepidoptera, Hymenoptera, Hemiptera, Diptera, Coleoptera, and Orthoptera [9,19,20,21,22,23,24,25].

In this study, we explored *T. castaneum Iflavirus* (*TcIV*) aiming for the complete genome sequence and the organization of *TcIV*. We also show the prevalence, epidemiology, tissue distribution, phylogeny, and transmission mechanism of *TcIV*.

## 2. Materials and Methods

### 2.1. The Insects

*T. castaneum* strains used in this study are listed in Appendix A. The populations we used were from colonies reared in different laboratories before the experiments were performed. The Goliath strain was the main population used for the experiments in this study that have been reared in our laboratory for more than 10 years. The insects were maintained at 28 °C in Golden Buffalo organic wheat flour mixed with 5% Brewer’s yeast (w/w) in glass jars [26].

### 2.2. Cell Culture

We used BCIRL-TcA-CLG1 (TcA) cell line derived from *T. castaneum* adult and pupae. Cells were reared in T25 flasks (25 cm^2^ tissue culture flasks version with the VENT screw cap, TPP Cat # 90025) with 3–5 mL of Ex-CELL 420 Serum-Free Medium for Insect Cells (Sigma Aldrich, Cat # 14420C, Burlington, Massachusetts) containing L-glutamine supplemented with 5% fetal bovine serum (FBS) and kept at 28 °C. Fetal bovine serum (FBS) was obtained from Atlas biologicals (Fort Collins, CO, USA). The cells were passaged on a weekly basis with half of the medium being replaced by fresh medium every 7–10 days [3,26]. The contamination of the cells by *TcIV* was confirmed by reverse transcription quantitative PCR (RT-qPCR).

### 2.3. SRA Data Search

The NCBI search tool was used to access all SRA (Sequence Read Archive) data available for *T. castaneum* as of March 2021 (https://www.ncbi.nlm.nih.gov/sra/?term=tribolium+castaneum) (accessed on 15 March 2021). From 1496 SRX for *T. castaneum*, 924 SRX transcriptomic sequences were used in this study (Appendix A). The Blast search (Basic Local Alignment Search Tool) in NCBI by using the query sequence of AUE23905.1 was followed by a back blast to confirm that the match was truly *TcIV* specific. The number of aligned sequences to show in the blast search was set to a maximum of 5000. The SRX samples with hits were further analyzed using the analysis tab for the presence of the *TcIV,* and the transcript per million value (TPM) was calculated by dividing the number of BLASTN hits by the number of total spots for each of the SRX data. Those without hits were excluded from further analysis [27].

### 2.4. RT-qPCR Assays

To study the prevalence of *TcIV* in the current *T. castaneum* colonies, we employed RT-qPCR to test forty-two *T. castaneum* lines and six closely related Tenebrionid species from different laboratories. Ten adults, approximately one month old from each strain, were randomly selected and pooled for total RNA extraction. Total RNA was extracted using a phenol-chloroform extraction method (Trizol reagent) (Zymo Research, Cat # R2050-1-200, Irvine, CA) following the manufacturer’s instructions. The RNA quality was evaluated by electrophoresis in 1% agarose gel, and the quantity was determined on a NanoDrop 2000 Spectrometer at 260 nm (Thermo Scientific, Wilmington, DE, USA) [3].

In this study, the first and second strand cDNA in all multiplex RT-qPCR assays was synthesized using primers and probes designed for the amplification of the RNA-dependent RNA polymerase (*RdRp*) region of the polyprotein with the Luna Universal Probe One-Step RT-PCR kit (New England Biolabs, Cat # E3006L, Ipswich, MA, USA). Each 10 µL reaction contained 5 µL of Luna Universal Probe One-Step Reaction Mix (2X), 0.5 µL of Luna WarmStart^®^ RT Enzyme Mix (20X), 0.4 µL of RdRp-Forward primer, 0.4 µL of RdRp-Reverse primer, 0.2 µL of RdRp-Probe, 0.4 µL of rpS3- Forward primer, 0.4 µL of rpS3- Reverse primer, 0.2 µL of rpS3-Probe (all at 10 µM concentration), and template RNA < 1 µg. The mRNA level of *RdRp* was measured with thermocycling at the following conditions: reverse transcription at 55 °C for 10 min, initial denaturation at 95 °C for 1 min, denaturation at 95 °C for 10 s, and extension at 60 °C for 30 s. The RT-qPCR was set up at 0.1 mL × 96 semi-skirted Fast Type PCR Plate A1 Cut, Natural (MidSci, # PR-PCR2196F), sealed with Adhesive Sealing Sheets (Thermo Scientific, Cat # AB-0558, Waltham, MA), and Real-time PCR amplification, and analysis was performed on a Bio-Rad CFX Connect Real-Time System and analyzed with CFX Manager^TM^ Software Version 3.1 (Bio-Rad Laboratories, Inc. CA, USA). The mRNA expression level was measured using the comparative CT (cycle threshold) method (2^−ΔΔCt^) by using the *rpS3* as the reference gene. The oligonucleotides used in this study are listed in Appendix A. All multiplex RT-qPCR assays were performed in three replications [28]. GraphPad Prism version 9.2.0 (GraphPad Software Inc., La Jolla, CA, USA) was used for all statistical analyses.

### 2.5. TcIV Genome Annotation and Phylogenetic Analysis

The open reading frame (ORF) sequence of *TcIV* submitted by Lewis et al. in 2018 [9] was accessed from NCBI (GenBank: AUE23905.1). The 5′ and 3′ untranslated region (UTR) sequences of the genome were generated using our EST (expressed sequence tagged) data from *T. castaneum* [29]. In order to capture the conserved functional motifs, the full amino acid sequence of *TcIV* was aligned with that of the deformed wing virus [30], slow bee paralysis virus (SBPV) [15], and *Drosophila suzukii* La Jolla virus [19] by using SnapGene software version 6.0.7.

Molecular evolutionary analyses were conducted using MEGA11 [31]. Iflavirus sequences (24 reference sequences), available in NCBI as of May 2022, as well as other iflaviruses, were selected for phylogenetic analyses using polyprotein Aimelvirus 1 [32], polyprotein Rabovirus D1 [33], and polyprotein Tottorivirus A1 [34] from family Picornaviridae as the outgroup. The full amino acid alignment was made using the MUSCLE alignment tool with the default settings and the phylogenetic tree was constructed using the Maximum Likelihood method with bootstrap values set to 500 replicates. Initial tree(s) for the heuristic search were obtained automatically by applying Neighbor-Join and BioNJ algorithms to a matrix of pairwise distances estimated using the JTT model and then selecting the topology with superior log likelihood value. This analysis involved 51 amino acid sequences. There was a total of 4557 positions in the final dataset.

### 2.6. TcIV in Different Tissues and in Vertical Transmission

Five randomly selected female and male Goliath beetles (~one-month-old in Park laboratory) were used for a tissue distribution assay using RT-qPCR. The individuals were first rinsed with sterile water to remove any attached flour and then dissected in PBS (phosphate buffered saline) (Accuris Instruments, Cat #EB1200, Edison, NJ, USA) under a stereomicroscope using sterile tweezers. The heads including the brain, gut, gonads, ganglia, and remaining carcass from each individual, were dissected and kept separately in a Trizol reagent at −80 °C for subsequent RNA extraction. The tweezers were heat-sterilized using a flame after dissecting each tissue, and the remaining parts were transferred to a fresh PBS drop to reduce and eliminate virus cross-contamination as much as possible. Total RNA was extracted and served as a template for the multiplex RT-qPCR assay as described previously. We dissected fat bodies from three additional individuals. The housekeeping control gene used was *rps3,* and the data were normalized using the 2^−ΔΔCt^ method and the Ct value of the head from each individual as a control [28].

In order to test whether *TcIV* transovarial transmission occurs, we collected eggs laid overnight by Goliath females and washed them with sterile water to remove any attached flour in a chamber made with 200 µm mesh screening. Eggs were then surface-sterilized by dipping the chamber in fresh 1% sodium hypochlorite (NaOCl) solution [21] and moved up and down for one min (2x) in order to soak the eggs thoroughly. The eggs were rinsed with sterile water in between two washes in 1% NaOCl in order to remove any remaining flour and external viruses. A total of 50 eggs for each of the three biological replications was transferred to a Trizol reagent for RT-qPCR assay in G_0_ eggs. After sterilization, the remaining eggs were lined up on a covered glass and transferred separately to fresh or contaminated flour until they hatched and began to grow. The contaminated flour was prepared from the flour inhabited by a Goliath colony for more than one month. The early instar larvae (G_0_), hatched from 1% NaOCl-sterilized eggs, in fresh and contaminated flour, were collected, and five randomly selected individuals were pooled for total RNA extraction and RT-qPCR, with three replications. The larvae that hatched from non-sterilized eggs in fresh flour were used as a control.

### 2.7. Immunocytochemistry (ICC)

In this study, we used the TcA cell line derived from *T. castaneum* adult and pupa, for ICC assays. Cells were seeded in 100 µL of the cell suspension in each well of a µ-Slide 18 Well Glass Bottom plate (ibidi, Cat # 81817) and placed in a Petri dish with a wet paper towel in order to provide the necessary humidity and cultured overnight at 28 °C. The following day, the medium was removed from each well and replaced with 100 µL of 2% paraformaldehyde (Electron Microscopy Sciences, Cat # 15960-01, Hatfield, PA) as a fixative for 4 h. After fixation, the cells were washed 3X with PBS + 0.5% Triton X-100 (PBST) (Accuris Instruments, Cat # EB1201, Edison, NJ) for about 2 h and then treated with 5% normal goat serum (NGS) prepared in PBST for 1 h. The primary antibody (Ab), produced in Rabbit (GenScript) and affinity purified with the epitope peptide in VP2 (INLFEWTTASTQGAL, shown in Figure 1), was diluted at 1:100 (Primary Ab: 5% NGS) and incubated with the fixed cells overnight. The cells were washed 3X with PBST for 4 h. Then the secondary Ab (FluoroNanogold^TM^-anti rabbit Fab’ Alexa Fluor^®^) (NanoProbes, Cat# 7204, Yaphank, NY) 1:500, was added to the wells overnight. The next day, the cells were washed with PBST more than three times. For imaging, the nuclei were stained with Hoechst (Invitrogen, Cat # H21492, Waltham, Massachusetts) at 2.5 µg/mL (in 5% NGS) for 20 min and scanned using LSM700 confocal microscopy. All incubations for the ICC experiment were done at room temperature.

## 3. Results

### 3.1. TcIV Genome Organization

We report the full-length genome sequence of *T. castaneum iflavirus* (GenBank accession # AUE23905.1), which is the assembly of our transcriptomic data [29], adding the 5′ and 3′ UTR sequences to the previous sequence reported by Lewis et al., 2018 [9] (GenBank accession #OP874944). We show that *TcIV* with a 10,158 bp RNA genome, has a 767 bp long 5′ UTR, and 192 bp 3′ UTR, tailed with a poly-A region. DWV, an extensively studied iflavirus, was found to be closely related to *TcIV* with 33.96% amino acid sequence identity and was used for the prediction of the genome organization of *TcIV* and the proteolytic cleavage sites (PCS). The structural proteins, located at the N-terminus, include a leader protein, 963 bp, (cleavage at ^318^ALPE321MD^323^), followed by the viral capsid proteins, with the order of VP2, 747 bp or 28.4 kDa (cleavage at ^567^PYPE570MD^572^), VP4, 63 bp or 2.4 kDa (cleavage at ^588^DNNR591DN^593^), VP3, 1248 bp or 46.7 kDa (cleavage at ^1004^AYAE1007GE^1009^), and VP1 with 786 bp or 44.2 kDa (cleavage at ^1398^VSPE1401AL^1403^) (Figure 1). We also predicted the non-structural proteins in the C-terminal part of the genome with the order of helicase (cleavage at PCS6: ^1875^GVPE1878MD^1880^), 3C-protease (cleavage at PCS7: ^2483^LFRE2486MF^2488^) and *RdRp*. The locations and sequence of the antibody used in ICC and the nucleotide length of each predicted region are shown in Figure 1. We compared the proteolytic cleavage sites (PCS1-5) of *TcIV* with that of DWV, SBPV, and DsLJV [18,19,30,35] (Appendix A). The amino acid numbers in the cleavage sites are based on their position in the amino acid sequence of each virus (Table 1). The amino acid motifs for cleavages are generally well conserved with the DWV sequence. The cleavage between PE and MD in PCS1 and 2 are strictly conserved while other cleavage motifs contain moderate levels of divergence.

### 3.2. TcIV Prevalence

In this study, we evaluated 43 different strains and 7 different non-*T. castaneum* Tenebrionid species to detect and measure the prevalence of *TcIV* by using qPCR. Samples were collected from laboratory colonies in March 2021. Ten individuals (about one-month old) from each strain were randomly collected and pooled for virus detection assays. *TcIV*-positive strains showed varying levels of *TcIV* in the ranges of 13.8 to 37 for the Ct (Figure 2). The beetles with no amplifications within 40 cycles of PCR were also common in *T. castaneum* populations: USDA [26], Columbia-1, Japan-4, Brazil-4, Z-2, Z-4, Hudson-KS [30], Abilene-1 [36], NDG-2, Costa Rica-1, Brazil, Thailand, CO-Pyr-R, Brazil-5, Shellenberger, A20 Rdiel, and in other Tenebrionid species: Belgium (*T. confusum*), *T. freemani*, *T. brevicornis*, *Palorus ratzeburgii*, *Cynaeus angustus*, *Gnathocerus cornutus*, and *Latheticus oryzae* (Appendix A).

### 3.3. TcIV Phylogenetic Analysis

To construct the phylogenetic tree, we compared the complete amino acid sequence of *TcIV* with 48 iflaviruses accessed from NCBI on March 2021. The DWV virus is found to be closely clustered with *TcIV*, with 33.96% amino acid identity [30] (Figure 3). A Hubei coleopteran virus 1 [37] was found to be closely related to the sequence we report here as *TcIV* with 90% amino acid sequence identity. The branches with taxa belonging to the same order were compressed and the name of the order and the accession numbers of the compressed taxa are shown in the tree. The clustering pattern of *TcIV* was identical in other methods, neighbor joining tree, and maximum parsimony tree.

### 3.4. TcIV in Sequence Read Archive Data (SRA)

We found SRA data that reveal a prevalence of *TcIV* among different *T. castaneum* populations grown on different continents. We obtained 924 SRX samples and performed reciprocal BLASTN searches using BLASTN against the *TcIV* full nucleotide sequence as an initial query against each SRX. We further analyzed the SRX samples for the number of matcheshit numbers (set to 5000 max hits) and the percent of the total reads number for each SRX. The data from the same BioSample were grouped together for presenting the data in Table 2. The SRA accession data for samples that did not contain reads derived from free-of-*TcIV* can be found in Appendix A for complete SRA information).

### 3.5. Tissue-Specific Distribution of TcIV

In order to map the *TcIV* tissue distribution in *T. castaneum*, we randomly collected five female and five male Goliath beetles (~one-month-old) and dissected the previously mentioned tissues for *TcIV* detection with RT-qPCR. A separate set of data for the fat body was obtained from three additional individuals. The Ct value from the head (Ct = ~20) was used as a control to normalize the data with the reference gene rpS3. We found that *TcIV* is distributed in all tested tissues with higher levels in ganglia in both females and males (Figure 4) and significantly lower levels in the gut and gonads (*p* ≤ 0.0001).

### 3.6. Vertical Transmission of TcIV

We evaluated whether the *TcIV* is transovarially transmitted. The eggs laid overnight by Goliath females were collected, and after washing off the attached flour with distilled water, they were sterilized with 1% NaOCl until dechorionation and removal of the virus particle from the egg’s surface. Our results show that the surface-sterilization procedure significantly decreased the level of the *RdRp* gene by 63% (paired *t*-test, *p* = 0.0161) (Figure 5a). The larvae that hatched from the sterilized eggs were kept in either fresh or contaminated flour, which originated from rearing the Goliath strain. The data show that the mRNA level of *RdRp* was similar in larvae hatched from sterilized eggs in both virus-free and contaminated flour (one-way ANOVA: F = 0.8817, df = 7, *p* = 0.4699) (Figure 5b).

### 3.7. TcIV Infections in TcA Cell

In order to visualize the *TcIV* in the TcA cell line, we first confirmed the presence of the virus in the cells using RT-qPCR. In immunocytochemistry, we observed two types of staining in our TcA cell line: staining in extracellular debris (Figure 6b), which is more prevalent and rarely in intracellular particles (Figure 6c). Unfortunately, we were unable to visualize the virus puncta in beetle tissues due to background interferences. In addition, we found the cells undergo the formation of mineral crystals in the cell culture as a diagnostic of *TcIV* contamination in the cell line (Figure 6d–f).

## 4. Discussion

Our results provide a general description of *T. castaneum iflavirus* in our laboratory population of the Goliath line. Due to the absence of symptomatic infection in our beetle colonies, we first learned about the iflavirus in *T. castaneum* when it was reported by Lewis et al. in 2018 [9]. They found small interfering RNAs targeting iflavirus in *T. castaneum,* likely as an immune mechanism against the *TcIV*. We confirmed the presence of iflavirus in our multiple colonies of *T. castaneum* and TcA cell line by RNA-seq and PCR-based strategy [21].

Iflaviruses are arthropod-specific viruses that infect insects from the insect orders Lepidoptera, Hymenoptera, Hemiptera, Diptera, Coleoptera, and Orthoptera [10]. Recently, Juergens et al., 2022 [27], identified two novel iflaviruses named the King virus and Rolda virus from bat guano in Washington state. Since these bats were captive and fed solely on *Tenebrio* spp., the insect diet of the bat might have been the source of the virus in them. In the surveys of *TcIV* in 43 populations of *T. castaneum* and 7 populations of non-*T. castaneum* species, we found that *TcIV* is highly specific only in *T. castaneum* and prevalent in different geographic strains from different laboratories. This survey of Tenebrionidae included *T. confusum* and *T. freemani*, which are closely related species to the *T. castaneum* but was limited to only one strain of each of these two species and will need to be expanded to other populations to make a more solid conclusion about these two species (Figure 2).

Even though there are some reports of lethal infections by iflaviruses in insects, such as pupation failure in honeybee larvae infected with sacbrood virus [38] and premature death in honeybees carrying deformed wing virus [17], such infections commonly appear to be covert and symptomless [10,14,39,40,41]. Iflaviruses often show no signs of overt infection in a number of insect species [39,40,41,42] and are usually detected through next-generation sequencing [27,43]. Often, they are found to cause sublethal effects on their hosts. Reduced larval weight and proportions of females were reported in *Spodoptera exigua* larvae infected by *S. exigua* iflavirus 1 (SeIV1) [40]. Declining honey bee colonies are associated with the negative effects of iflavirus on foraging adult workers, and the survival of the honey bees by DWV was also reported [14]. In *Helicoverpa armigera,* the larvae infected by the *H. armigera* iflavirus (HaIV) showed significant changes in the gene expression levels of the host [41]. Iflaviruses can also play significant roles in the host’s susceptibility to lethal infections by other pathogens. For example, covert infection by *S. exigua* iflavirus 1, induced by prior inoculation of egg masses, increased mortality and nucleopolyhedrovirus (SeMNPV) infection rates in that species [39,40]. We found different degrees of infections in different strains of beetles from different laboratories, which showed that the infection rate and degree were highly variable. A number of strains showed no detectable levels of infection which provide an opportunity to further investigate strain-specific interactions between the *TcIV* and *T. castaneum*.

Mineral crystals have been reported in cell cultures derived from cat kidney cortex and urinary bladders infected with a feline syncytium-forming virus and another feline virus that produced intranuclear inclusions [44]. The mineral crystals we observed in our TcA cell line may serve as a diagnostic tool for high levels of *TcIV* infection in the cell line.

So far, available data indicate punctuated distribution and phylogeny of iflaviruses over host insect taxa with local exceptions at the Order level. In our evolutionary analysis, *TcIV* was closely clustered with iflaviruses from Hymenoptera, Lepidoptera, and Ixodida, confirming the observation of Silva et al., 2015 [25], that iflaviruses do not generate a single clade according to the infected order. This implies that they do not follow the same evolutionary path as their insect hosts at the order level (Figure 3). However, the clusters of some iflaviruses within Lepidoptera, Hymenoptera, and Hemiptera indicate that the host-iflavirus association could have been ancestral to these taxonomic groups, although horizontal transmission within the taxonomic groups might have also occurred.

Both horizontal and vertical transmission of iflaviruses have been reported [10]. The high load of the virus would be expected to be a symptomatic infection and increases the probability of horizontal transmission through the production of contaminated feces and regurgitation [45]. Vertical transmission, however, is usually accompanied by low levels of virus and symptomless infections [10]. We conclude that the asymptomatic infection of *T. castaneum* by *TcIV*, via both mechanisms of vertical and horizontal transmission, seems to be the transmission pathway based on our experimental results for surface sterilization of eggs that retained *TcIV* at 37% compared to the unwashed control eggs (Figure 5a).

Virus distribution in host tissues varies depending on the host species. HaIV (*H. armigera* iflavirus) [21] and HaDV2 (*H. armigera* densovirus 2) [46] were mainly found in the fat body. Sacbrood virus in the honey bee was also found mainly in the fat body [47,48]. We found that *TcIV* is in all tested tissues with high levels in the nervous tissues (ganglia) and low levels in the gut and gonads (Figure 4). Whether the tissue-specific infection is associated with symptomatic infection is unknown.

In conclusion, *TcIV* is widespread among laboratory cultures of *T. castaneum*, including the strains that were recently collected from the field. It covertly infects its host without detectable pathogenicity and is transmitted through both vertical and horizontal pathways. *TcIV* offers an opportunity to study how and under what conditions this virus interacts with the immune system in *T. castaneum*, a model beetle species.

## Figures and Tables

**Figure 1 insects-14-00220-f001:**
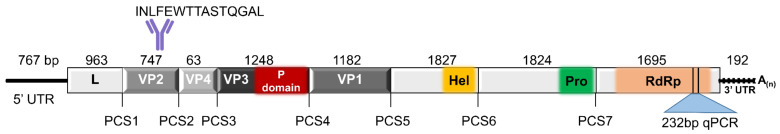
*T. castaneum Iflavirus* genome annotation based on the full amino acid sequence aligned with DWV: Deformed Wing Virus [30]; SBPV: Slow Bee Paralysis Virus [35] and DsLJV: *Drosophila suzukii* La Jolla Virus [19]. UTR: untranslated region; L: leader protein; VP2, 4, 3 and 1: viral capsid proteins 2, 4, 3 and 1; Hel: helicase; Pro: 3C-protease; *RdRp*: RNA dependent RNA polymerase; PCS: protease cleavage site.

**Figure 2 insects-14-00220-f002:**
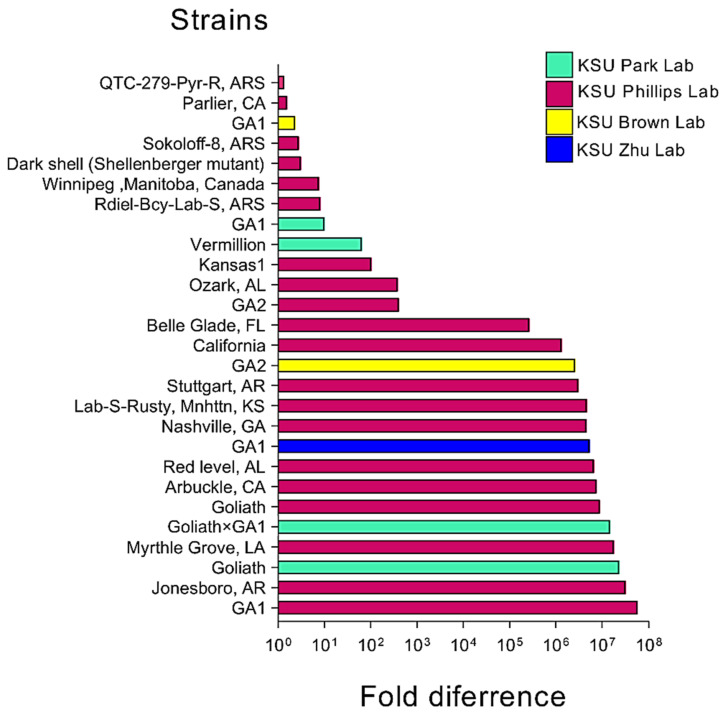
*TcIV* prevalence measured by *RdRp* qPCR in different *T. castaneum* strains. QTC-279-Pyr-R strain having the Ct cycle 37 was used as reference 1. Please note that the colors indicate different laboratories that had the same strains with different levels of *TcIV*. Details for the strains are in Appendix A.

**Figure 3 insects-14-00220-f003:**
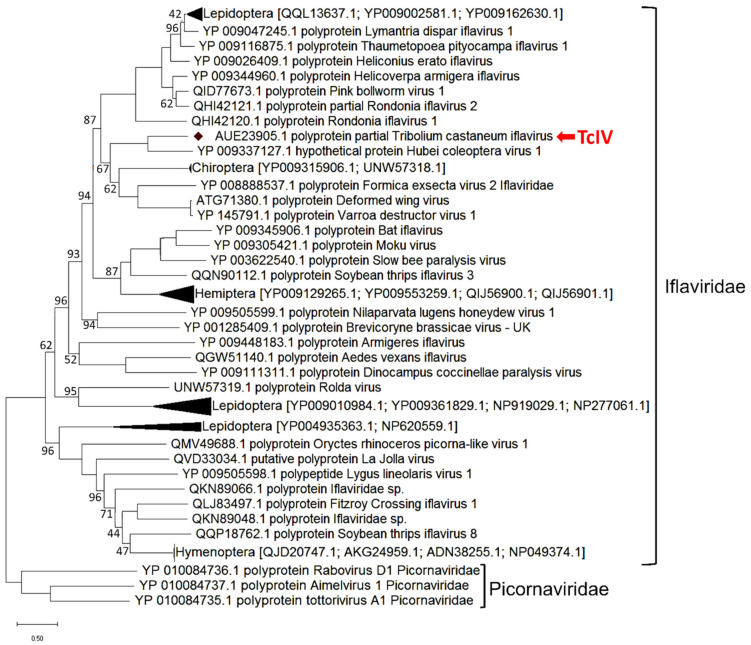
Evolutionary tree for *TcIV* with other members of iflaviridae. The tree was constructed using the Maximum Likelihood method with the JTT matrix model in MEGA based on a MUSCLE alignment. Three strains from family Picornaviridae were used as an outgroup. Bootstrapping values on each node represent the percent in 500 replicates. The nodes without value represent a bootstrap of 100. The black triangles represent the compressed branches with multiple taxa belonging to the same Order.

**Figure 4 insects-14-00220-f004:**
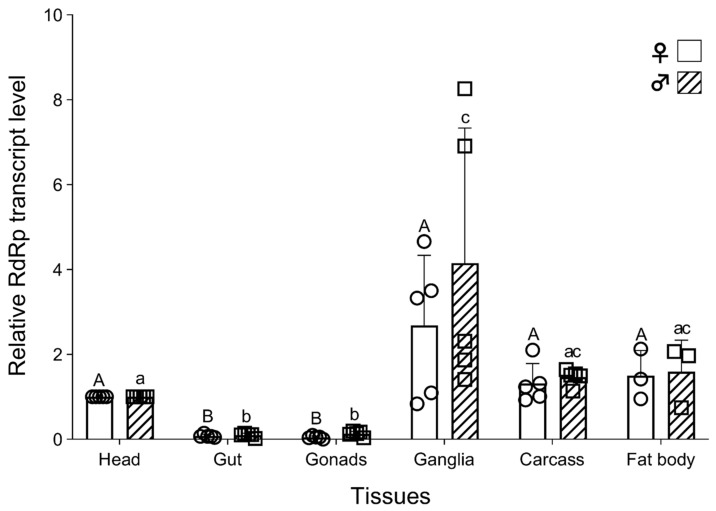
The levels of *RdRp* in different tissues of female and male Goliath *T. castaneum*. Data were normalized using the 2^−ΔΔCt^ method using the head from each individual as a control with the rpS3 of each sample. The bars represent the mean, and the circles and squares are the raw data points for every single tissue derived from females and males, respectively. Standard deviation of each bar is shown for n = 5 (for fat body n = 3). Letters on the top of the bars are significant differences at *p* = 0.05 in the ANOVA test. Upper case is for the female data and lower case is for the male data.

**Figure 5 insects-14-00220-f005:**
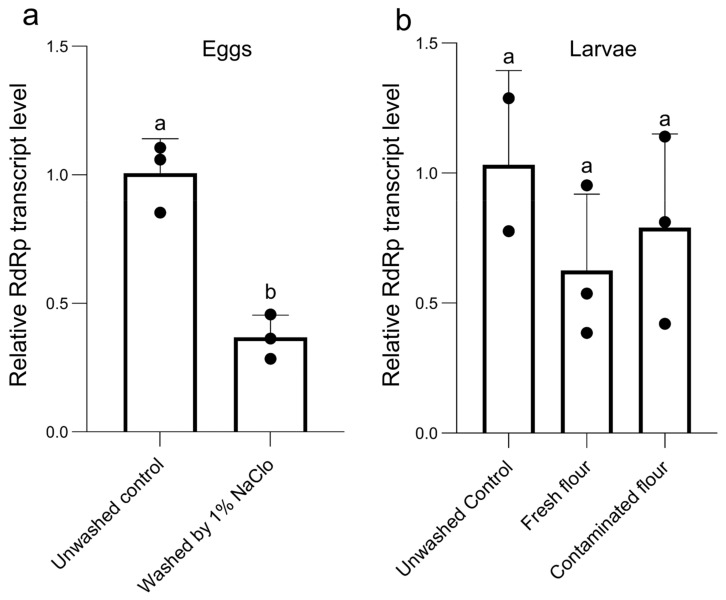
The mRNA levels of *RdRp* from eggs washed by 1% NaOCl (**a**) the larvae hatched from those eggs in fresh or contaminated flour (**b**). Data were normalized using the 2^−ΔΔCt^ method. The bars represent the mean of three replications (each data point represents one replication) and the standard deviation of each bar is shown for n = 3.

**Figure 6 insects-14-00220-f006:**
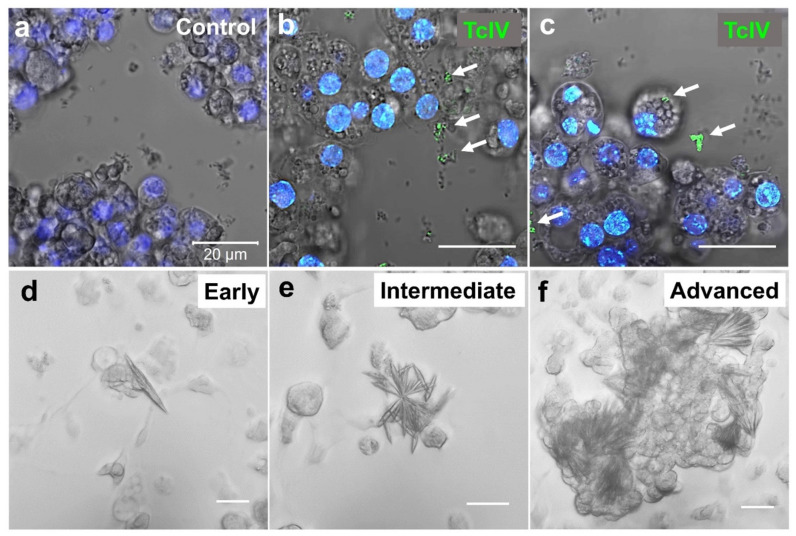
TcA cell line immunocytochemistry (**a**–**c**) and mineral crystal formations (**d**–**f**). TcA cell line was stained with an antibody raised in rabbit against *TcIV*-VP2. Rabbit IgG control did not show specific staining (**a**). The green dots pointed by white arrows show the virus particles stained with *TcIV* antibody in extracellular debris (**b**) and intracellular particles (**c**). The blue staining shows the nuclei stained with Hoechst (2.5 µg/mL). (**d**–**f**) show the different degrees of mineral crystal formation in TcA cell line infected by *TcIV*. Image processing was done by ZEN Microscopy Software. The bars are all for 20 µm.

**Table 1 insects-14-00220-t001:** Comparison of putative capsid proteins cleavage sites in different iflaviruses (*TcIV*: *T. castaneum iflavirus* [9]; DWV: Deformed Wing Virus [30]; SBPV: Slow Bee Paralysis Virus [35] and DsLJV: *Drosophila suzukii* La Jolla Virus [19]. The numbers are based on the putative translations of each virus.

	Conserved Protease Cleavage Sites in Different Iflaviruses				
	PCS1	PCS2	PCS3	PCS4	PCS5
	LP; VP2	VP2; VP4	VP4; VP3	VP3; VP1	VP1; C-Terminus
*TcIV*	ALPE321MD	PYPE570MD	DNNR591DN	AYAE1007GE	VSPE1401AL
DWV	AKPE211MD	AKPE464MD	GNNM485DN	AIPE901GE	AVPE1288AP
SBPV	AQPE176MD	ALPE437MD	DVNC458DN	AEPE888ME	AMPE1303GF
DsLJV	PQPQ171ME	PVVQ408ME	IKNM432DK	AFAQ856MD	PSSQ1113ME

**Table 2 insects-14-00220-t002:** The summary of SRA data containing *TcIV* ranked from low to high. The percent transcript is based on the hit numbers in BLASTN search and NCBI categorization.

SRX Accession #	Geographic Location	Strain	Total # of Reads (Range)	% Transcripts
SRX1336374..SRX1381752	USDA ARS	Unknown	1.06~19.81 M	0~0.015%
SRX1381239..SRX1433712	USDA ARS	Unknown	2.72~36.36 M	0~0.07%
SRX2591944..SRX2591991	Vanderbilt University (GEO)	Unknown	8.99~24.77 M	0.0002~0.05%
SRX390599..SRX391110	University of Muenster	SB/Cro1	21.82~42.24 M	0.0005~0.02%
SRX2695346	USDA ARS left for Grain and Animal Health Research	Oppert lab	154.78 M	0.003%
SRX818546..SRX819651	Wayne State University	GAI	23.79~66.43 M	0.005~0.02%
SRX3119712..SRX3118448	Wayne State University	Unknown	43.02~88.05 M	0.006~0.06%
SRX7097353..SRX7140859	Jiangsu University	Unknown	63.4 M	0~0.01%
SRX2390214.. SRX2390215	USDA ARS	Unknown	31.40~32.62 M	0.015%
SRX2265569..SRX2265586	University of Muenster	Cro1	19.60~29.83 M	0.02%
SRX1925245..SRX1925248	Nanjing University (GEO)	GA-1	5.85~6.11 M	0.02~0.08%
SRX5103179..SRX5103190	South China Agriculture University	Unknown	7.89~10.38 M	0.05~0.06%
SRX3406391	Nanjing Normal University	Unknown	6.85 M	0.07%
SRX5361486..SRX5361487	Nanjing Normal University	Unknown	70.59~74.60 M	0.07%
SRX2944938	Nanjing Normal University	GA-1	6.83 M	0.07%
SRX1513481	Nanjing Normal University	GA-1	7.31 M	0.07%
SRX476140-SRX699016	Nanjing Normal University	GA-1	7.12~7.58 M	0.07%
SRX1081871	Nanjing Normal University	GA-1	6.96 M	0.07%
SRX6408778..SRX6408837	University of Münster (GEO)	Cro1	13.28~21.57 M	0.01~93.19%
SRX6808996..SRX6812159	Harvard University	Unknown	7.52~46.48 M	0.12~0.59%
SRX8875633..SRX8875638	Yangzhou University	GA-1	24.7~25.97 M	0.19~70.67%

## Data Availability

The full length TcIV sequence is available in the GenBank with accession #OP874944.

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
