# Peer review of "Characterization of Iflavirus in the Red Flour Beetle, Tribolium castaneum (Coleoptera; Tenebrionidae)"

_insects, 2023, doi:10.3390/insects14030220_

Round 1
Reviewer 1 Report
The authors confirmed that the presence of iflavirus in multiple colonies of T. castaneum and TcA cell line using RNA-seq, PCR based strategy and ICC. It covertly infects its host without detectable pathogenicity and is transmitted through both vertical and horizontal pathways. The data are convincing and the manuscript is well written. I only have a few minor issues for improvement.
1. The authors used the affinity-purified polyclonal antibody to detect the presence of iflavirus in TcA cell lines. It will certainly improve the quality of the manuscript if the iflavirus is detected in one of the infected tissues such as CNS since the iflavirus was widespread in multiple colonies of T. castaneum.
2. It seems to me the authors did not mention the specificity of polyclonal antibody used in the study. It will be nice to have a WB showing the antibody specificity.
3. P5 L194: Figure 1 caption “T. castaneum Iflavirus genome annotation”. Please italicize T. castaneum.
4. P5 L195: “Drosophila suzukii La Jolla Virus”. Please italicize Drosophila suzukii.
5. P6L206: Italicize “T. castaneum”.
6. P6L219: Italicize “T. castaneum” in Figure 2 caption.
7. P8L255: “P=<0.0001”. Please change P to p and italicize p.
8. P8L257: Italicize “T. castaneum” in Figure 4 caption.
9. P9L261, 268, 272: Italicize p.
The authors confirmed that the presence of iflavirus in multiple colonies of T. castaneum and TcA cell line using RNA-seq, PCR based strategy and ICC. It covertly infects its host without detectable pathogenicity and is transmitted through both vertical and horizontal pathways. The data are convincing and the manuscript is well written. I only have a few minor issues for improvement.
1. The authors used the affinity-purified polyclonal antibody to detect the presence of iflavirus in TcA cell lines. It will certainly improve the quality of the manuscript if the iflavirus is detected in one of the infected tissues such as CNS since the iflavirus was widespread in multiple colonies of T. castaneum.
2. It seems to me the authors did not mention the specificity of polyclonal antibody used in the study. It will be nice to have a WB showing the antibody specificity.
3. P5 L194: Figure 1 caption “T. castaneum Iflavirus genome annotation”. Please italicize T. castaneum.
4. P5 L195: “Drosophila suzukii La Jolla Virus”. Please italicize Drosophila suzukii.
5. P6L206: Italicize “T. castaneum”.
6. P6L219: Italicize “T. castaneum” in Figure 2 caption.
7. P8L255: “P=<0.0001”. Please change P to p and italicize p.
8. P8L257: Italicize “T. castaneum” in Figure 4 caption.
9. P9L261, 268, 272: Italicize p.
Reviewer 2 Report
This is a well-written manuscript that reports the presence of an Iflavirus in the Red Flour Beetle and its derived cell line TcA. This manuscript covers all essentials to report a virus sequence and also measured the infection level in different lab populations. I only have some minor comments which I think will improve the quality of the manuscript.
Minors:
ICTV roles should be followed carefully for writing the virus names. please check the entire of manuscript.
T. castaneum Iflavirus (TcIV), this is not right: full name should be italic if this is a species Tribolium castaneum iflasvirus (all italic) otherwise Tribolium castaneum iflasvirus (not italic)
Page 2 line 59: please cite the reference for the TcA cell line here.
line 71: which tool did you use for BLAST?
line 74: why 5000? Any justification
line 112-113 what was the aim of this alignment? describe, please. Have you done this for genome annotation? have you done any domain searches (i.e. Pfam & CDD)?
Line 119: phylogenetic analysis requires more description which model did you use? have you done any tests to find the best fit for the model?
It is not clear if they used all polyprotein sequence or only specific domains like RdRP.
Line 219: T. castaneum should be italic. Check the other places too.
Line 224: I assumed the polyprotein sequences have been used for this phylogenetic analysis. You have the full genome sequence of this virus why did you use the partial polyprotein sequence for TcIV? See your figure 3
Line 233: are they real strains?
Table 2: the % of Transcripts is too low in a few data sets and it is good to include the total number of reads in the library and TcIV-derived reads to have a better view.
Figure 4: it is a relative transcript level, not an actual RdRp mRNA level
Line 304: are you sure they have been isolated? or they just have been reported by RNA-seq data analysis!
Line 335-336: I do not agree with this statement. Mineral formation in the cell culture media can happen due to many factors and it is not easy to say it is happing because of virus infection. No one expects to see them in a live insect. Re-consider this statement.
Author Response
Please see attached point-by-point response
